# Precarious Job Makes Me Withdraw? The Role of Job Insecurity and Negative Affect

**DOI:** 10.3390/ijerph182412999

**Published:** 2021-12-09

**Authors:** Shanting Zheng, Tangli Ding, Hao Chen, Yunhong Wu, Wenjing Cai

**Affiliations:** 1School of Tourism and Events, Hefei University, Hefei 230601, China; zhengst@hfuu.edu.cn; 2School of Management, Fudan University, Shanghai 200433, China; Tangli1123@outlook.com; 3College of Economics & Management, Anhui Agricultural University, Hefei 230036, China; chenhao0508@163.com; 4School of Public Affairs, University of Science and Technology of China, Hefei 230026, China; wyhhappy1234567@mail.ustc.edu.cn; 5Department of Management & Organization, Vrije Universiteit Amsterdam, 1081 HV Amsterdam, The Netherlands; 6Intellectual Property Research Institute, University of Science and Technology of China, Hefei 230026, China

**Keywords:** job precariousness, job insecurity, withdrawal behavior, negative affect

## Abstract

An expanding “gig” economy has changed the nature of employment; thus, researchers have recently focused on exploring the role of job precariousness in the workplace. However, little research attention has been given to understanding why, how and when job precariousness leads to employees’ negative behavioral outcomes in the service-oriented industry. In the current study, we examined job insecurity as a mediator and employees’ negative affect as a moderator in the relationship between job precariousness and employees’ withdrawal behavior. Using a sample of 472 employees working in Chinese hotels, we found that job precariousness is positively related to employees’ withdrawal behavior by increasing their job insecurity. Moreover, this mediating relationship is conditional on the moderator variable of employees’ negative affect for the path from job insecurity to withdrawal behavior. The importance of these findings for understanding the undesirable behavior outcomes of job precariousness is discussed.

“Workers are forced to bear the risk of any downturn in the employer’s business because workers can be disposed of immediately when they’re no longer needed.”—International Labor Organization

## 1. Introduction

In recent decades, significant changes have swept the global labor market, particularly the proliferation of precarious work [1,2]. Vosko [3] defines precarious work as a paid job characterized by uncertainty, low income and limited social benefits and statutory entitlements. Typical, contingent and nonstandard work, such as agency, part-time, fixed-term, temporary, casual, seasonal and on-call work, are examples of precarious work [4]. The ILO [5] notes that temporary employment, a kind of precarious employment, has increased steadily in Organization for Economic Co-operation and Development (OECD) countries since the 1980s. People engaged in precarious work are particularly vulnerable to economic disruptions. In the past, firms adopted precarious forms of work arrangements to reduce costs and increase flexibilities [6]; flexible labor resources were especially prevalent in the service industry because they allowed a firm to adjust its capacity to match supply to current service demand [6,7].

The crux of the aforementioned problem lies not only in objectively precarious work itself but also in people’s psychological experience of subjective job precariousness [8]. In fact, individuals’ subjective perceptions of external conditions are often better predictors of outcomes than objective conditions because people’s expectations of the work play an important role [9]. For example, those with temporary work may have lower expectations, which mediates the relationship between objective and subjective job insecurity [10]. Therefore, subjective perceptions exert more proximal effects on outcomes. Empirical studies have demonstrated over and again that subjective job insecurity could be more aggravating than actual job loss or dismissal [9,11]. In addition, job precariousness accounts for individual differences in the sensitivity to and capability of coping with stressors [12], while precarious work is a description of job characteristics that do not involve individual differences. Although precarious work has been extensively studied, job precariousness has received limited scholarly attention. Only recently have some scholars begun to separate precarious work from job precariousness to explore the downstream outcomes of job precariousness. For instance, Vives, et al. [13] find that job precariousness is positively associated with poor mental health, and Mai et al. [14] conclude that objectively precarious working conditions undermine sleep by promoting subjective job precariousness. However, these empirical studies document the adverse effects of job precariousness on physical and psychological health, significantly overlooking its impacts on workplace behavioral outcomes. Because behavioral outcomes are practically important and universal in real work settings, linking them to precarious jobs would enrich our understanding of a range of organizational phenomena [12].

Relating to our research context of the service industry, the interaction between a frontline service provider and a customer is a key feature that distinguishes service activities from manufacturing [15], and the worker–customer interface is a significant factor that shapes customer experience. Therefore, the workplace behavior of service providers is extremely important in the service industry. Little is known, however, regarding how service providers behave when they face omnipresent job precariousness. Previous studies assume that when individuals perceive unfavorable job characteristics, they tend to respond with withdrawal behavior [16,17] and coping strategies; dissatisfied individuals seek to avoid such work situations [18]. Withdrawal behavior refers to a set of attitudes and behaviors used by employees when they maintain a job but, for some reason, attempt to be less participative [19]. It represents a sequence of behaviors that begins with episodic psychological withdrawal and, through increasing degrees of withdrawal, gradually expands to physical withdrawal [16]. Accordingly, do employees withdraw from work when they experience precariousness? To the best of our knowledge, research on the effect of job precariousness on workplace behaviors, such as withdrawal behavior, is nonexistent, as are studies of the underlying mechanisms and contextual moderators. Accordingly, this paper aims to fill these research gaps by answering the following question: How and when does job precariousness affect employee withdrawal behaviors in a service setting?

To understand whether job precariousness engenders employees’ withdrawal behavior, we draw on conservation of resources (COR) theory, which suggests that individuals strive to obtain, retain, foster and protect those things that they centrally value, while the principle of primacy of resource loss states that it is more harmful to lose resources than to gain resources [20,21]. Therefore, when losing or holding few resources, individuals will endeavor to restore their resources. In line with this perspective, we argue that job precariousness creates a feeling of resource scarcity and ensuing stresses, which in turn stimulates one’s desire to protect and restore one’s resources. In particular, we use employees’ perception of job insecurity as a key mediator to comprehensively understand how one’s present perception of resource scarcity results in a continuing perception of resource shortages in the future [22]. Moreover, given the detrimental effects of withdrawal behavior on most occasions, another important question worth exploring concerns the boundary conditions that may exacerbate negative effects—that is, some moderating factors that organizations could identify to prevent negative outcomes. To this end, we identify the role of negative affect as demonstrated in previous studies and examine its moderating effects. Figure 1 shows the theorized model in the current study.

This paper contributes to the extant literature in several ways. First, drawing on COR theory [20], this study sheds light on how employees react to their perception of job precariousness with certain behaviors. Previous studies have documented the negative impacts of job precariousness on health [23,24,25], but limited research has been conducted to scrutinize employees’ coping behaviors. We fill this gap by studying the behavioral consequences of job precariousness in the workplace. Meanwhile, we contribute to the literature on job insecurity by enriching the current understanding of its antecedents, given that the existing research examining experiential factors (e.g., perceived employability, emotional exhaustion) as antecedents of job insecurity has primarily focused on the threat of a lack of personal resources, overlooking a resource shortage induced by a job per se [22]. Second, by examining the indirect effects of job precariousness on withdrawal behavior via job insecurity, we provide empirical evidence that a sense of resource scarcity will extend from one’s present to the future, generating an ongoing perception of job insecurity, which is then responded to via one’s withdrawal behavior. Finally, by identifying the enhancing effect of negative affect [26], the current study not only depicts individual differences in withdrawal behavior but also offers practical implications for organizations regarding how to select employees and how to care for employees with a high negative affect who are subject to precarious work arrangements.

## 2. Theoretical Background and Hypotheses Development

### 2.1. Job Precariousness

Job precariousness has been theoretically conceptualized within two related streams. The first is a combination of objective job features—that is, a paid job characterized by uncertainty, low income and limited social benefits and statutory entitlements [3]. The second is a subjective experience of precarious work involving the individual, psychological experience of precariousness related to one’s work [8,14]. Consistent with previous studies, in the current study, we follow the second view, defining job precariousness as an individual’s subjective perception of objective job features because an individual’s subjective perception of external conditions has been suggested to be a better predictor of outcomes than objective perceptions [12]. Conceptually, as a perceptual construct, job precariousness is defined and measured differently in existing studies [8,14]. We followed previous studies that have evidenced that the job precariousness scale (JPS) developed by Creed, Hood, Selenko and Bagley [8] has more reliability and validity to adopt their conceptualization and measurement in the current study.

It is worth noting that job precariousness is different from job insecurity in definition, as job insecurity is the perception of potential job loss as a whole, as well as a deterioration in job conditions and a loss of anticipated opportunities [22]. Although they are both conceptual, job precariousness is a phenomenon that is present-oriented, implying the perception of holding a low-quality job [27], while job insecurity is future-oriented, indicating the perception of potential job-threatening factors [22]. When developing a job precariousness scale, Creed, Hood, Selenko and Bagley [8] also provided support for construct validity, in which job precariousness and job insecurity are different constructs.

The consequences of job precariousness most often documented are the deterioration of physical and mental health on all kinds of indicators [28,29,30]. A meta-analysis conducted by Quinlan and Bohle [31] revealed that more than 80% of articles showed a negative association between precarious work and workplace wellbeing (e.g., physical health and safety). In addition, job precariousness is considered a chronic work stressor and, unsurprisingly, has a negative impact on occupational attitudes such as job satisfaction [32], organizational trust [33] and job commitment [34]. However, due to the lack of effective measurement, empirical studies on the impact of job precariousness on workplace behavior outcomes are very rare.

### 2.2. Job Precariousness and Withdrawal Behavior

Job precariousness is generally an unpleasant work experience [35], whereby the perception that one’s job is uncertain, with a low rate of pay and limited or no social benefits and statutory entitlements, is a significant stressor for individuals [14,36]. When individuals experience a high level of job precariousness, they tend to feel a sense of discomfort that undermines their sleep habits [14] and leads to more financial strain and less life satisfaction [8]. Thus, job precariousness is a perception of resource scarcity at work and can lead to further resource depletion. According to COR theory, which defines resources as things that one values—specifically, objects, states and conditions [37]—resource gains, such as job certainty, income level, social benefits and statutory entitlements, are all valued resources. However, employees with precarious jobs have few of these valued resources. COR theory suggests that individuals with more resources tend to gain more resources, while individuals with fewer resources are likely to experience resource losses [21]. Thus, employees with precarious jobs are more likely to experience resource losses, which will drive employees into certain levels of stress [21,38].

Drawing on the resource-gaining and resource-losing arguments of COR, we expect that when employees experience job precariousness in their workplace, they are likely to display disengaged activities. Specifically, when aiming to alleviate the stress incurred by job precariousness, employees tend to establish defensive attempts to conserve their remaining resources, to recover from losses and to gain resources [37]. One way to restore resources at work is to withdraw from the workplace (e.g., absenteeism, tardiness, turnover) [39]. On the basis of COR theory, we argue that when an employee is occupied with a precarious job, he or she faces high stress due to resource shortages and may use withdrawal behavior as a coping mechanism to restore resources [40]. Specifically, employees who put less effort into their job (slowing the pace of resource drain from an unsupportive work environment) or take longer lunches or rest breaks (retaining and building more resources via a relaxing environment) might restore their lost resources. As such, we propose the following:

**Hypothesis** **1** **(H1).**
*Job precariousness is positively related to withdrawal behavior.*


### 2.3. The Mediating Role of Job Insecurity

Furthermore, we propose the specific psychological process by which job precariousness leads to withdrawal behavior by arguing that job precariousness triggers job insecurity among employees, which in turn causes withdrawal behavior. Shoss [22] defines job insecurity as a phenomenon that is perceptual, future-oriented and uncertain; it can refer to the potential loss of a job as a whole, as well as a deterioration in job conditions and a loss of anticipated opportunities. When employees experience high levels of job precariousness, they face great uncertainty regarding their future work, suffer from low income and receive limited social benefits and statutory entitlements [3]. Therefore, they are in a state of job resource shortage. According to COR theory, stress occurs due to an insufficient amount of resources required to complete the job at hand [37]. This lack of resources and the ensuing stress act as a threat to one’s current job; because the level of precariousness will not improve in the foreseeable future, employees are likely to foresee a dim future and experience a sense of job insecurity. Therefore, stress caused by a lack of job resources in the present results in worry about the potential loss of job-related resources in the future [22]. Thus, employees’ current perception of job precariousness leads to their future-oriented perception of job insecurity.

How does job insecurity trigger withdrawal behavior? The broad mechanism of this trigger includes a two-stage stress appraisal process [41,42]. In the primary stage, employees identify whether a situation could have a significant impact on their wellbeing; in the secondary stage, they establish coping strategies to reduce stress and improve their wellbeing [43]. Yi and Wang [44] argue that job insecurity could be a hindrance stressor or a challenge stressor, depending on how one appraises a stressor and on how many resources a person has to cope with the stress it produces. In the context of employees who experience job precariousness, job insecurity is a hindrance stressor because a job offers insufficient resources for employees to cope with stress [45]. When job insecurity is appraised in this way, people usually disengage from their job and organization, displaying less positive work attitudes and behaviors [46]. For instance, they may choose to withdraw from their workplace [43]. COR theory suggests that both a perceived and an actual loss of resources can induce further psychological stress [47], and people who report reduced resources are more likely to withdraw from their workplace to supplement their depleted resources [48]. As such, we propose the following:

**Hypothesis** **2** **(H2).**
*Job insecurity mediates the positive relationship between job precariousness and withdrawal behavior.*


### 2.4. The Moderating Role of Negative Affect

According to the theoretical argumentation in the framework of COR theory, positive personality traits, such as optimism, self-efficacy and positive affect, are personal, characteristic resources that help individuals to resist and cope with stress [21,49]. In contrast, a negative affect entails that one possesses a low level of personal, characteristic resources. Researchers have suggested that individuals with a high negative affect experience more negative emotions because they perceive the world more negatively than individuals with a low negative affect [50]. We follow previous studies and suggest that such personal, emotion-related characteristics may play a contingent role in the relationship between job precariousness and withdrawal behavior via job insecurity. Negative affect, or negative affectivity, is a personality trait that involves the experience of negative emotions and a poor self-concept [51]. Individuals with a high negative affect are characterized as excessively sensitive to minor frustrations and irritations and are more likely to experience a variety of negative emotions, such as anxiety, anger, contempt, disgust, guilt, fear and nervousness [52]; in contrast, people with a high positive affect are characterized by frequent states of calmness and serenity and tend to exude confidence, activeness and great enthusiasm.

When confronted by stressful conditions, including the perception of precariousness-induced job insecurity that threatens one’s job and job features, employees with a high negative affect are rather likely to experience helplessness, sadness and fear [53]. Trapped by their whirls of negative emotions and negative self-concepts, they have insufficient resources to aggressively confront stress, resorting instead to withdrawal behavior to protect themselves from being harmed by their stress and to restore their depleted resources [21]. On the other hand, employees with a low negative affect may face job insecurity in a healthier fashion; they tend to have stronger problem-solving abilities [54]. They are able to take a psychological break or create a respite from stress [55], which can replenish the resources depleted by their stress [56]. Therefore, when the stress associated with job insecurity can be buffered by more mature copying styles, employees tend to resort to withdrawal behaviors to restore their lost resources much less often. As such, we propose the following:

**Hypothesis** **3** **(H3).**
*Negative affect moderates the indirect relationship between job precariousness and withdrawal behavior through job insecurity such that the indirect relationship becomes stronger for employees with higher negative affect.*


## 3. Methods

### 3.1. Sample and Procedure

We used survey questionnaires as the research design to collect data in the service industry. Specifically, we invited employees working in hotels in China to participate in our research. Before submitting our questionnaires, we received a list of local hotels in a city in the middle area of China from the governmental organization (i.e., Department of Cultural and Tourism of A Province). There were more than 150 hotels in the list. We first randomly selected ten hotels by asking about their willingness to participate in our research. This random selection process could create representative samples from which conclusions could be drawn and applied to the larger population. After receiving confirmation of participation from four hotels, we contacted the HR departments in each hotel to ask for their assistance. These HR departments asked all their employees in the hotels, providing the aims and descriptions of our current research project, to confirm whether they would participate. Then, the HR departments provided a list of employees (*n* = 753) in their hotels who were willing to join our research project by completing our questionnaires at two different times. In the list, the HR departments reported the six numbers of these participants’ ID cards and their telephone number for our research team’s ease of identification. These participants were informed that their responses would be anonymous and kept confidential; that is, they only reported six numbers of their ID cards to help researchers to match their responses from the two different times. Since we used a time-lagged research design, one of the authors submitted paper-and-pencil questionnaires to these employees during their working hours at two different time points, with a time interval of one month. Specifically, one of the authors in our research, who was an assistant in the process of submitting questionnaires in these hotels, contacted these participants during their working hours by calling them to fill in the questionnaires. After completing these questionnaires, these employees reported their surveys back to the author directly. At Time 1, 753 employees were asked to rate their job precariousness, job insecurity and negative affect. Meanwhile, they were also required to provide their personal demographic information. We received 625 responses and finally obtained 591 usable responses after deleting the incomplete responses in accordance with previous related studies [57]. One month later, at Time 2, we submitted our questionnaires to the 591 employees, who were asked to rate their own withdrawal behavior, and received 492 responses. After deleting incomplete responses, we had 472 valid responses.

Among the participants, most were female (67.4%), and their average age was 29.24 years. The most frequently indicated education level was an institute of technology degree (55.5%). Their average working tenure in the current organization was 2.40 years.

### 3.2. Measures

Since all the scales were originally from English versions, the translation–back translation procedure was employed to translate all the scales from English to Chinese [58]. Respondents rated each of the items on a 5-point scale from 1 = strongly disagree to 7 = strongly agree, unless noted otherwise.

#### 3.2.1. Job Precariousness

We used a scale from Creed, Hood, Selenko and Bagley [8] to measure employees’ perception of job precariousness. The entire scale consisted of 12 items, 3 of which, referring to the dimension of job insecurity, were excluded. Thus, the final scale used in the current study included 3 dimensions: job conditions (3 items), job remuneration (3 items) and job flexibility (3 items). Sample items for the three dimensions were “Are you able to negotiate working conditions that better suit you? (job conditions)”, “Does your pay meet unexpected expenses? (job remuneration)” and “Are you able to take time off if you are unwell without worrying about losing your job or being penalized (e.g., hours cut)? (job flexibility)”. Respondents rated each of the items on a 6-point scale from 1 = not at all to 6 = a great extent. The Cronbach’s α was 0.88.

#### 3.2.2. Job Insecurity

We followed Jung et al. [59] and used an eight-item scale to assess job insecurity. Employees were asked to evaluate their overall perceptions of job insecurity. A sample item was “I feel uncertain about the future of my job.” The Cronbach’s α was 0.97.

#### 3.2.3. Negative Affect

A ten-item scale from Watson, Clark and Tellegen’s (1988) Positive and Negative Affect Schedule (PANAS) was used to measure employees’ state affect. Sample items included “upset” [60] and “distressed”. The Cronbach’s α was 0.96.

#### 3.2.4. Withdrawal Behavior

We asked employees to self-rate their own withdrawal behavior in the workplace with 11 items from Lehman and Simpson [16]. Sample items for psychological withdrawal behavior and physical withdrawal behavior were “How often have you thought of being absent” and “How often have you left work early without permission”, respectively. Respondents rated each of the items on a 7-point scale from 1 = never to 7 = very often. The Cronbach’s α was 0.99.

#### 3.2.5. Control Variables

We controlled the following variables: gender (1 = female; 2 = male), age (in years), educational level (1 = high school, 2 = institute of technology, 3 = bachelor’s degree, 4 = master’s degree and above) and working tenure in the current organization (in years).

## 4. Results

### 4.1. Validity Analyses

Before testing our hypotheses, we conducted a series of analyses to establish the validity. First, due to the one-source data set (i.e., from employee ratings), we tested for the presence of common method bias (CMB). Specifically, we performed Harman’s single factor test with principal axis factoring (PAF) as an extraction method to examine whether most of the variance could be explained by a single factor [61,62]. The results revealed multiple distinct factors, with the first unrotated factor accounting for only 39.09% of the total variance extracted. Thus, CMB was not a serious concern in our data. Second, we conducted a confirmatory factor analysis (CFA) for job precariousness, job insecurity, negative affect and withdrawal behavior to evaluate the discriminant validity of the measure in the present research. As shown in Table 1, alternative models indicated a poor fit to the date—i.e., the three-factor model (χ^2^/*df* = 10,864.66/662.00, *p* < 0.001, CFI = 0.69, RMSEA = 0.18, IFI = 0.69, TLI = 0.67, RMR = 0.16), the two-factor model (χ^2^/*df* = 15,072.68/664.00, *p* < 0.001, CFI = 0.56, RMSEA = 0.22, IFI = 0.56, TLI = 0.53, RMR = 0.13) and the one-factor model (χ^2^/*df* = 19,581.45/665.00, *p* < 0.001, CFI = 0.42, RMSEA = 0.25, IFI = 0.42, TLI = 0.39, RMR = 0.22). Our proposed four-factor model demonstrated a better fit to the data (χ^2^/*df* = 2963.86/623.00, *p* < 0.001, CFI = 0.93, RMSEA = 0.08, IFI = 0.93, TLI = 0.92, RMR = 0.06) than all the alternative models. Therefore, the results provided strong support for the distinctiveness of the four study variables for subsequent analyses [63].

Table 2 presents the descriptive statistics, reliabilities and correlations of all the variables in the current study. Consistent with our expectations, job precariousness was positively correlated with job insecurity (*r* = 0.45, *p* < 0.01) and withdrawal behavior (*r* = 0.63, *p* < 0.01). Moreover, job insecurity was positively correlated with withdrawal behavior (*r* = 0.55, *p* < 0.01).

### 4.2. Hypothesis Testing

To test mediation effects, we used SPSS 26.0 to test the hypotheses using hierarchical regression analyses. To further clarify the mediation effect, we employed SPSS PROCESS Model 4 using a bootstrap procedure with 5000 samples to produce a confidence interval (CI) for the indirect effect. In Table 3, the result indicates that after controlling for the effect of employees’ gender, age, education and tenure, job precariousness was positively associated with withdrawal behavior (*β* = 0.63, *p* < 0.001), thus supporting H1. In addition, job precariousness has a positive effect on job insecurity (*β* = 0.47, *p* < 0.001), and job insecurity has a positive effect on withdrawal behavior (*β* = 0.33, *p* < 0.001). After job precariousness and job insecurity were both entered into Model 3, the results showed that the relation between job precariousness and withdrawal behavior was reduced (*β* = 0.48, *p* < 0.001). Regarding the mediating effect, we tested the significance of the indirect effect using the bootstrapping technique [64]. Specifically, as Table 4 shows, the bootstrapped confidence interval (95% CI [0.12; 0.27]) did not include zero. Therefore, the mediating effect was significant, supporting H2.

We then tested the moderated mediation model by using SPSS PROCESS Model 14. As shown in Table 5, the indirect effect of job precariousness on withdrawal behavior through job insecurity was moderated by negative affect (*β* = 0.11, *p* < 0.001). The results of the bias-corrected confidence intervals in Table 6 show that the indirect relation is significant both when negative affect is higher (indirect effect = 0.28, 95% CI [0.18; 0.38]) and when negative affect is lower (indirect effect = 0.13, 95% CI [0.07; 0.23]). The index of moderated mediation was 0.05 (SE = 0.02, 95% CI [0.005; 0.083]), thus supporting H3. To illustrate more clearly the role of negative affect, we plotted the simple slope test in Figure 2, which shows that negative affect significantly strengthens the relation between employees’ job insecurity and withdrawal behavior.

## 5. Discussion

To explore how (through job insecurity) and when (high negative affect) job precariousness may contribute to employees’ withdrawal behavior, drawing on COR theory, we theoretically proposed and empirically tested a positive relationship between job precariousness and employees’ withdrawal behavior, both psychologically and physically. This relationship was mediated by employees’ perception of job insecurity. Moreover, our survey findings indicated that the indirect relationship between job precariousness and withdrawal behavior through employees’ job insecurity was significant when employees’ negative affect was high rather than low. Therefore, the results suggest that when employees experience negative affect in the workplace, their perception of job precariousness is more likely to lead to undesirable outcomes such as withdrawing from their work by increasing their job insecurity.

### 5.1. Theoretical Implications

The present study has some theoretical implications for the job precariousness literature. First, we empirically advance research on the behavioral-related outcomes associated with job precariousness [65,66]. Previous studies have particularly evidenced that when individuals perceive job precariousness, they tend to have negative experiences (e.g., physical and psychological ill health) [23,24]. However, existing evidence fails to illustrate the potential influences of job precariousness on employees’ work-related behaviors, especially negative behaviors, although scholars have theoretically indicated that working in precarious work settings may potentially stimulate individuals to act out certain behaviors associated with the context [8]. Therefore, our findings enrich the theoretical claim by empirically showing that employees with precarious experiences are likely to engage in withdrawal behaviors. In this vein, we are unique in moving beyond these findings of undesirable behavioral outcomes to scrutinize employees’ coping behaviors. Therefore, we respond to the need for an in-depth understanding of why job precariousness relates to behavioral-related outcomes in the workplace [66,67].

Second, we integrate the job precariousness literature with core features from COR theory on resource loss in the workplace to identify employees’ job insecurity as an intervening process mechanism that can link employees’ perception of job precariousness to their withdrawal behavior. In doing so, we address a recent call to draw on diverse theories to provide insights into the job precariousness literature [12]. Specifically, employing COR theory, we theoretically highlight the “resource” perspective of precarious jobs to explain its impact on employees’ corresponding responses. In other words, shedding light on the theoretical assumption that precarious work is uncertainly related to personal loss of valuable resources [35], we theoretically indicate that job precariousness triggers individual psychological stress when valued resources are under threat of loss or actually lost; therefore, they face the threat of job insecurity. Moreover, our examinations of the indirect effects of job precariousness on withdrawal behavior via job insecurity further suggest that a sense of resource scarcity definitely extends from one’s present into the future, generating an ongoing perception of job insecurity and thus responding with one’s withdrawal behavior.

Our findings also contribute to the current understanding of the antecedents of job insecurity. In particular, past research primarily focused on examining the threat of a lack of personal resources toward individuals’ sense of job insecurity, ignoring a resource shortage induced by a job per se [22]. In this way, we not only empirically distinguish the constructs of job precariousness and job insecurity [68] but also link the association between them from the resource approach. In other words, precarious work acts as a threatening resource to increase individuals’ powerlessness, anxiety and social isolation related to work [69], which in turn leads to an increase in insecurity at work.

Third, the results illustrate that how employees experience negative affect moderates the mediating process mechanism of job insecurity in the job precariousness–withdrawal behavior relationship. More specifically, employees who experience job precariousness are more likely to sense insecure jobs and then display withdrawal behavior in the workplace when they have a high level of negative affect. In other words, personal affect can act as a critical condition for job precariousness to foster withdrawal behavior. This is consistent with the theoretical framework of COR, suggesting that personality traits are personal resources that influence individuals’ coping strategies [21,49]. Moving beyond some well-examined contextual factors as moderators, our studies join a growing body of work showing that additional moderators can be identified in terms of personal characteristics (e.g., Allan et al., 2021). Future research might explore and identify additional boundary conditions in the indirect relationship between job precariousness and withdrawal behavior through job insecurity by considering employees’ positive affect to further examine whether positive resources may attenuate resource loss.

Finally, our results derived from the service industry can be generalized to other industries. Scholars studying employment trends have argued that “uncertainty and unpredictability, and to varying levels personal risk, have spread into a broad range of post-industrial workplaces, services and production alike” [70,71]. Moreover, some industries are characterized by more precariousness than others, such as construction (bogus self-employment), agriculture (seasonal work) and food processing (fixed-term work) [72,73]. Therefore, our results can also shed light on the need for other industries to pay more attention to job precariousness.

### 5.2. Practical Implications

According to the findings in the present research, some practical implications can be highlighted to help organizations, especially service-oriented organizations (e.g., hotels), in decreasing employees’ negative outcomes. First, given the evidence of the harmful effect of job precariousness among employees, organizations should first properly identify who is permanent and who is precarious. For example, when recruiting candidates, a test of precariousness can be provided to them. This would help organizations to properly select employees. Because service-oriented industries are more vulnerable to a “gig” economy, employees are more likely to experience job precariousness. Thus, first and foremost, organizations should avoid misusing their power to deprive employees of their rights; instead, they should provide employees with due social benefits and statutory entitlements. Human resources departments should provide some training courses to help employees to not only develop skills and knowledge to cope with unstable employment but also receive mental health treatment and assistance with seeking job opportunities. Second, to decrease their feelings of job insecurity, employees are encouraged to identify factors that they do not have control over and do the best they can with the available resources. For example, they should be fully aware that, given the increasingly turbulent environment facing many modern companies, it is hardly possible for them to completely eliminate job insecurity. However, they can focus on what is within their grasp, such as constant learning and upskilling to maintain and increase their employability, in addition to better communicating their challenges with their managers and striving for more decision-making opportunities [74].

Finally, given the results on the moderating role of negative affect, employees should take the initiative to avoid their own negative attitudes by self-regulating their behaviors and attitudes. For example, they can change the path of emotions before they are fully experienced by selectively attending to aspects of the environment that elicit desired emotions. At the same time, organizations can engage in building a sound working environment where interpersonal relationships are trustful. For example, leaders could set an agenda to give employees performance feedback regularly and provide support for those in need [75]. In doing so, employees are more likely to bring optimism throughout the workplace. Additionally, organizations can use psychometrically sound personality measures to assess critical attributions in hiring and screening processes, as well as in employee care, because certain traits impact positive or negative motivational states and behaviors [76]. Our findings suggest the importance of screening for negative affect to provide targeted care, especially for those who experience high levels of job precariousness.

### 5.3. Limitations

Some limitations should be noted in the current research. The first limitation regards the one-source data collection. Although the analyses show that CMB is not a problem in our study, self-reported data can have pitfalls in our research context. For example, employees may over-evaluate their own withdrawal behavior; as a result, the objectivity of the results would be decreased. Future research is highly encouraged to include others’ ratings—that is, the assessments of colleagues, supervisors and customers may be of interest. Second, despite the time-lagged research design, we may have the problem of hypothesized causality because our independent variable, mediator and moderator were all measured at the same time (Time 1). Specifically, the sense of job insecurity in the workplace may induce employees’ experience of job precariousness. Therefore, it is highly recommended for scholars in the future to conduct longitudinal studies to replicate our research findings. Moreover, given that we claim the importance of investigating job precariousness among employees in the service industry, studies in the future should extend our research by inviting employees from other servicer-oriented organizations rather than hotels. In this vein, the findings would be established and generalized with more validity and reliability.

In addition, since precarious work reflects not merely temporary features of the business cycle but structural transformations of contemporary employment [27], job precariousness could be a chronic workplace stressor. Therefore, future research could take a stress perspective and examine the long-term effect of job precariousness on other behavior outcomes. Another limitation regards the sampling in the current study. Given that we employed the random sampling technique to collect data, we assumed no differences in samples between those who remained and those who dropped out of the study. Future research is highly recommended to take the sample who dropped out of the study into consideration for such data analyses as attrition analyses. Relatedly, we also encouraged scholars to adopt a more rigorous research design to replicate and re-establish our research findings.

Finally, although we followed COR to propose and test the contribution of job insecurity to employees’ withdrawal behavior, it is possible that vulnerable workers may try to make their jobs more secure by not engaging in withdrawal behavior. Thus, future research is encouraged to explore whether individuals with a certain type of personality would decrease their withdrawal behaviors when they feel the insecurity of their jobs. For example, employees with a proactive personality are more strategic in coping with difficulties and challenges. When they experience job insecurity at some point in the workplace, they are more likely to exhibit crafting behaviors to increase their social or structural job resources and challenging job demands and to decrease hindering job demands instead of generating withdrawal behavior.

## 6. Conclusions

In recent decades, precarious work has swept the global labor market, arousing interest in the psychological experience of job precariousness among employees. This paper sheds light on how and when job precariousness leads to employees’ negative behavioral outcomes (e.g., withdrawal behavior) in the service-oriented industry from the resource perspective. The results highlight the significant role of job precariousness not only as a constant workplace stressor but also as an antecedent of job insecurity, which in turn causes employees’ withdrawal behavior. Our results also show that withdrawal behavior is more likely to manifest in those with high rather than low negative affect, which should be considered when planning interventions at the individual level. Given the expansion of the “gig” economy and the deinstitutionalization of organized workers, we believe that job precariousness will continue to be a predominant employment, psychological and health issue and that research on this issue is of great importance and necessity.

## Figures and Tables

**Figure 1 ijerph-18-12999-f001:**
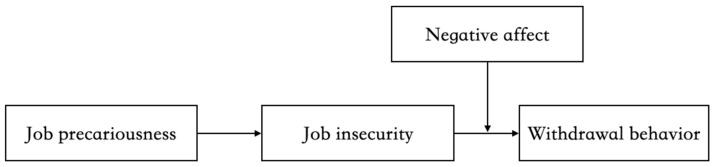
The hypothesized model.

**Figure 2 ijerph-18-12999-f002:**
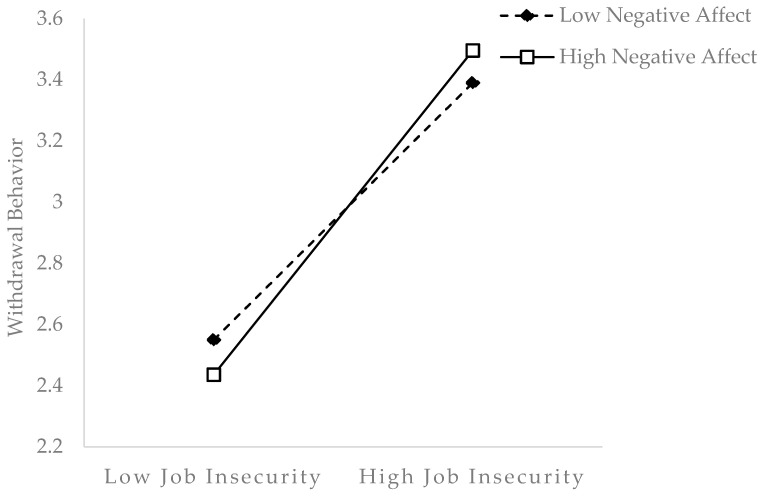
Negative affect as a moderator in the relationship between job insecurity and withdrawal behavior.

**Table 1 ijerph-18-12999-t001:** Confirmatory factor analysis.

Variables	χ^2^	*Df*	CFI	RMSEA	IFI	TLI	SRMR
Four-factor model	2963.86	623.00	0.93	0.08	0.93	0.92	0.06
Three-factor model	10,864.66	662.00	0.69	0.18	0.69	0.67	0.16
Two-factor model	15,072.68	664.00	0.56	0.22	0.56	0.53	0.13
One-factor model	19,581.45	665.00	0.42	0.25	0.42	0.39	0.22

*n* = 472; four-factor model: job precariousness, job insecurity, withdrawal behavior, negative affect; three-factor model: job precariousness + job insecurity, withdrawal behavior, negative affect; two-factor model: job precariousness + job insecurity + withdrawal behavior, negative affect; one-factor model: job precariousness + job insecurity + withdrawal behavior + negative affect.

**Table 2 ijerph-18-12999-t002:** Mean, standard deviations and correlations among the variables.

Variables	Mean	SD	1	2	3	4	5	6	7
1. Job precariousness	5.12	0.99							
2. Job insecurity	6.17	1.03	0.45 **						
3. Withdrawal behavior	5.86	1.20	0.63 **	0.55 **					
4. Negative affect	4.02	1.42	−0.08	0.04	0.002				
5. Gender	1.33	0.47	−0.01	0.02	0.004	0.09			
6. Education	1.97	0.68	−0.14 **	−0.04	−0.09 *	0.10 *	0.29 **		
7. Age	29.24	6.92	−0.09 *	0.04	−0.02	0.06	0.15 **	0.11 *	
8. Tenure	2.40	0.96	−0.05	−0.03	−0.07	0.06	0.10 *	0.05	0.42 **

*n* = 472; * *p* < 0.05, ** *p* < 0.01.

**Table 3 ijerph-18-12999-t003:** Results of the mediation effects of job insecurity.

Outcome Variable: Job Insecurity		Outcome Variable: Withdrawal Behavior	
	Model 1	Model 2	Model 3
Independent variable			
Job precariousness	0.47 ***	0.63 ***	0.48 ***
Mediator			
Job insecurity			0.33 ***
R^2^	0.22	0.40	0.49
ΔR^2^	0.21	0.39	0.09
F	25.49 ***	62.99 ***	73.99 ***

*n* = 472; *** *p* < 0.001.

**Table 4 ijerph-18-12999-t004:** Direct and indirect effects of job precariousness on withdrawal behavior.

**Direct Effect**			
Effect 0.58	SE 0.05	t 12.69	95%CI [0.49; 0.67]
**Indirect effect**			
Effect 0.19	Boot SE 0.04	Boot 95% CI [0.12; 0.27]	

*n* = 472.

**Table 5 ijerph-18-12999-t005:** Results of the moderating effects of negative affect.

	Outcome Variable: Job Insecurity				Outcome Variable: Withdrawal Behavior			
	Coefficients	SE	t	95%CI	Coefficients	SE	t	95%CI
**Control variables**								
Gender	0.01	0.10	0.05	[−0.18; 0.19]	0.01	0.09	0.08	[−0.17; 0.18]
Education	0.03	0.07	0.52	[−0.09; 0.16]	−0.03	0.06	−0.56	[−0.15; 0.09]
Age	0.01	0.01	2.14	[0.01; 0.03]	0.01	0.01	0.96	[−0.01; 0.02]
Tenure	−0.06	0.05	−1.22	[−0.15; 0.04]	−0.06	0.05	−1.32	[−0.15; 0.03]
**Independent variable**								
Job precariousness	0.48 ***	0.04	11.13	[0.40; 0.57]	0.57 ***	0.05	12.41	[0.48; 0.66]
**Mediator**								
Job insecurity					0.42 ***	0.04	9.45	[0.33; 0.51]
**Interactive effect**								
Job insecurity × Negative affect					0.11 ***	0.03	3.68	[0.05; 0.16]
Model summary	R	R^2^	MSE	F	R	R^2^	MSE	F
	0.46	0.21	0.84	25.49	0.71	0.50	0.72	58.74

*n* = 472; *** *p* < 0.001. All statistics were performed using PROCESS (Model 14) in SPSS.

**Table 6 ijerph-18-12999-t006:** Results of the moderated mediation.

Moderator Negative Affect	Indirect Effect	Boot SE	95% CI
Low levels of negative affect (−1 SD) High levels of negative affect (+1 SD)	0.13 0.28	0.04 0.05	[0.07; 0.23] [0.18; 0.38]

*n* = 472.

## Data Availability

The data of this study are available from the corresponding author upon request.

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
