# Peer review of "Precarious Job Makes Me Withdraw? The Role of Job Insecurity and Negative Affect"

_ijerph, 2021, doi:10.3390/ijerph182412999_

Round 1
Reviewer 1 Report
Precarious job makes me withdraw? The role of job insecurity and negative affect
The paper investigates how and when job precariousness affects employee’s withdrawal behaviour in a service industry setting. The paper needs some corrections and clarifications and is then ready to be published.
Minor issues
Line 46: the sentence should come with a qualification, i.e., why are subjective perceptions of external conditions better predictors than objective perceptions?
Line 70: Include your definition of withdrawal behaviour here (from line 151). Since this is in the title of the paper the definition should come early in the paper.
Line 74: your research question is to investigate how and when job precariousness affects employee’s withdrawal behaviour. This is stated differently in line 340 (“explore why and how…”) and in line 447 (“sheds light on how, why and when…”). Please chose one phrasing.
Line 235: you state that the sites for data collection were randomly selected. How was this done? What was the procedure for selecting the sites?
Line 245: did you do any analysis of missing values before deleting them? Could you detect a pattern in the missing values? Why did you choose to simply delete them and not to use an of the imputation techniques?
The Cronbach’s Alpha are quite high. This could be a sign of redundancy in the items.
Line 334: this sentence does not belong in the results section. Please delete or move it to the discussion section!
Line 347: this sentence is hard to understand. Please rewrite!
Author Response
RESPONSES TO REVIEWER 1’S COMMENTS
The paper investigates how and when job precariousness affects employee’s withdrawal behaviour in a service industry setting. The paper needs some corrections and clarifications and is then ready to be published.
Response: Thank you very much for your positive and encouraging comments on our manuscript. Your insightful guidance has helped us a lot during our revision. We have revised our paper in accordance with your comments and suggestions. We hope that you will find our revision addressing your concerns. Below are our point-by-point responses to your concerns.
Minor issues
Line 46: the sentence should come with a qualification, i.e., why are subjective perceptions of external conditions better predictors than objective perceptions?
Response: Thank you for this remind. This is a very important qualification and we added it in our manuscript. Specifically, we inserted that “In fact, individuals’ subjective perceptions of external conditions are often better predictors of outcomes than objective conditions because people’s expectations of the work play an important role [8]. For example, those with temporary work may have lower expectations, which mediates the relationship between objective and subjective job insecurity[9]. Therefore, subjective perceptions exert more proximal effects on outcomes. Empirical studies have demonstrated over and again that subjective job insecurity could be more aggravating than actual job loss or dismissal [8,10].” (p. 2).
Line 70: Include your definition of withdrawal behaviour here (from line 151). Since this is in the title of the paper the definition should come early in the paper.
Response: Thanks for your suggestion to include the definition of withdrawal behavior in the front part of the paper. We corrected this by theoretically emphasizing that withdrawal behavior represents a sequence of behaviors that begins with episodic psychological withdrawal and, through increasing degrees of withdrawal, gradually expands to physical withdrawal (Lehman & Simpson, 1992). (p. 2).
Line 74: your research question is to investigate how and when job precariousness affects employee’s withdrawal behaviour. This is stated differently in line 340 (“explore why and how…”) and in line 447 (“sheds light on how, why and when…”). Please chose one phrasing.
Response: Your suggestion is well taken and we modified all our statements to “how and when” to illustrate the mediating and moderating mechanisms. (p. 2).
Line 235: you state that the sites for data collection were randomly selected. How was this done? What was the procedure for selecting the sites?
Response: Thank you for this feedback. In the new manuscript, we added detailed information to reveal our sampling procedure. Specifically, we inserted that “Before submitting our questionnaires, we received a list of local hotels in a city in the middle area of China from the governmental organization (i.e., Department of Cultural and Tourism of A Province). There were more than 150 hotels in the list. We first randomly selected ten hotels by asking about their willingness to participate in our re-search. This random selection process could create representative samples from which conclusions can be drawn and applied to the larger population.” (p. 6).
Line 245: did you do any analysis of missing values before deleting them? Could you detect a pattern in the missing values? Why did you choose to simply delete them and not to use an of the imputation techniques?
Response: Thank you for this valuable comment. Following previous studies (Wang et al., 2014) , we used the listwise deletion approach to delete cases with missing data. This approach has been widely acknowledged and employed by researchers in management and psychology areas (cf. Peugh & Enders, 2004). Consistently, before submitting our questionnaires to the participants, we informed them that only employees who fill out surveys completely can be included in our research project. Thus, our valid response rate is relatively high.
In addition, acknowledging the significance of your point, we added a limitation to call for future studies that generalize to a target population including both full survey respondents and partial survey respondents. Please refer to the detailed argumentations on p.12.
The Cronbach’s Alpha are quite high. This could be a sign of redundancy in the items.
Response: Thank you for pointing this. As we mentioned in the text, all the measures were employed from previous studies which provided empirical evidence of high reliability and validity; thus, the high Cronbach’s Alpha are acceptable. Meanwhile, we referred the related papers which used the same scales to find that the Cronbach’s Alpha are all high (e.g., Scott & Barnes, 2011; Triana et al., 2017).
Moreover, acknowledging your points, we did some tests to analyze the data in order to support the high Cronbach’s Alpha. Specifically, we randomly deleted two items from each scale, and tested the reliability of the scales many times. After deleting any two items, the overall Cronbach’s Alpha of each scale is between 0.85-0.88 (the scale of job precariousness), 0. 96-0.97 (the scale of job insecurity), 0.99 (the scale of withdrawal behavior), 0.95-0.96(the scale of negative affect), indicating that all the items are necessary towards presenting a high Cronbach’s Alpha.
Line 334: this sentence does not belong in the results section. Please delete or move it to the discussion section!
Response: Thanks for this comment and we deleted this sentence “Therefore, employees should avoid having negative emotions, which is not conducive to their work” since it doesn’t belong to the results section.
Line 347: this sentence is hard to understand. Please rewrite!
Response: We accepted your advice by rewriting the sentence; that is “Therefore, the results suggest that when employees experience negative affect in the workplace, their perception of job precariousness is more likely to lead to undesirable outcomes such as withdrawing from their work by increasing their job insecurity.” (p. 10).

Reviewer 2 Report
This is an interesting and well-written piece of paper with a research focus on ‘gig’ economy. Specifically, the authors empirically examined why, how and when job precariousness leads to employees’ negative behavioral outcomes in the service-oriented industry. Overall, I would like to provide the following comments to further improve the paper:
- First of all, in the introduction section, I do not see a strong argument for exploring the downstream outcomes of job precariousness, especially relating job precariousness to workplace behavioral outcomes. Another point relates to the job precariousness in the service industry. Given its significance in making contributions theoretically, these argumentations (line 59-75) should be shown in the front.
- The authors proposed four contributions in the current study. However, as far as I sense, the second is similar to the first, which altogether enriches the job precariousness literature. Thus, I suggest the authors reframe the theoretical contributions in the current study. This would enhance the quality of your manuscript.
- In the sections of 2.1. Job Precariousness, the literature review on the links between job precariousness and certain outcomes from previous studies is lacking, although the authors have provided evidence on the conceptualizations of between job precariousness.
- This study collected data from the Chinese service-related sectors. The result of this study can be generalized into other sectors, which can be further described by the authors. Moreover, the author conducted a sampling way by submitting questionnaires to employees who can complete our questionnaires at Time 1 with the help of HR staff. However, how did you find these employees that are able to complete the questionnaires? This should be described clearer.
- The analysis was appropriate, sufficient and well-conducted. But I suggest that a report of the confirmatory factor analysis in a more thorough manner would also provide more strength to the current version.
- Under the practical implications section, the authors are suggested to provide literature to support the suggestions to the practical field. Moreover, specific cases to demonstrate the practices from the real world might be a good way to provide much clearer implications to readers.
- There are punctuation errors that hinder the quality of this manuscript. I recommend authors to go through the text with a revision perspective or seek aid from a professional editor.
Author Response
RESPONSES TO REVIEWER 2’S COMMENTS
This is an interesting and well-written piece of paper with a research focus on ‘gig’ economy. Specifically, the authors empirically examined why, how and when job precariousness leads to employees’ negative behavioral outcomes in the service-oriented industry. Overall, I would like to provide the following comments to further improve the paper:
Response: Thank you very much for your positive and encouraging comments on our manuscript. Your insightful guidance has helped us a lot during our revision. We have revised our paper in accordance with your comments and suggestions. We hope that you will find our revision addressing your concerns. Below are our point-by-point responses to your concerns
- First of all, in the introduction section, I do not see a strong argument for exploring the downstream outcomes of job precariousness, especially relating job precariousness to workplace behavioral outcomes. Another point relates to the job precariousness in the service industry. Given its significance in making contributions theoretically, these argumentations (line 59-75) should be shown in the front.
Response: Thank you for your insightful comments which have been taken as a major revision in our new manuscript. We did the following revisions:
- We added that “Because behavioral outcomes are practically important and universal in real work settings, linking them to precarious jobs would enrich our understanding of a range of organizational phenomena [11].” (p. 2, line 63-65). In this way, we hope to strengthen our research rationale and legitimacy.
- We moved our argumentations of our research context—i.e., service industry—to the front part of our introduction (i.e., in the end of the first paragraph, p. 1, line 40-43). In doing so, we aim to highlight our contribution of exploring job preciousness in the service industry.
- The authors proposed four contributions in the current study. However, as far as I sense, the second is similar to the first, which altogether enriches the job precariousness literature. Thus, I suggest the authors reframe the theoretical contributions in the current study. This would enhance the quality of your manuscript.
Response: Thank you for this constructive comment. In the new manuscript, we combined the first two contributions to specifically indicate that our current study fills the gap of precarious job scrutinizing employees’ coping behaviors by studying the behavioral consequences of job precariousness in the workplace.
- In the sections of 2.1. Job Precariousness, the literature review on the links between job precariousness and certain outcomes from previous studies is lacking, although the authors have provided evidence on the conceptualizations of between job precariousness.
Response: Thank you for pointing this out. It’s our omission not to review the consequences of job precariousness, so we added it in the last part of section 2.1. Specifically, we inserted “The consequences of job precariousness most often documented are deterioration of physical and mental health on all kinds of indicators [29-31]. A meta-analysis con-ducted by Quinlan and Bohle [32] revealed that more than 80% of articles showed a negative association between precarious work and workplace wellbeing (e.g., physical health and safety). In addition, job precariousness is considered a chronic work stressor and unsurprisingly has a negative impact on occupational attitudes such as job satisfaction [33], organizational trust [34] and job commitment [35]. However, due to the lack of effective measurement, empirical studies on the impact of job precariousness on workplace behavior outcomes are very rare.” (p. 3-4).
- This study collected data from the Chinese service-related sectors. The result of this study can be generalized into other sectors, which can be further described by the authors. Moreover, the author conducted a sampling way by submitting questionnaires to employees who can complete our questionnaires at Time 1 with the help of HR staff. However, how did you find these employees that are able to complete the questionnaires? This should be described clearer.
Response: Very grateful for your helpful suggestions. We revised our paper in accordance with your comments and suggestions. Below is our further description of the generalization of our results.
“Finally, our results derived from the service industry can be generalized to other industries. Scholars studying employment trends have argued that “uncertainty and unpredictability, and to varying levels personal risk, have spread into a broad range of post-industrial workplaces, services and production alike” [72,73]. Moreover, some industries are characterized by more precariousness than others, such as construction (bogus self-employment), agriculture (seasonal work) and food processing (fixed-term work) [74,75]. Therefore, our results can also shed light on the need for other industries to pay more attention to job precariousness.” (p. 11).
Moreover, in the sampling process section, we added more detailed contents to reveal how did we ensure employees to complete the questionnaires. Specifically, we inserted that “These HR departments asked all their employees in the hotels with aims and descriptions of our current research project to confirm whether they would participate. Then, the HR departments provided a list of employees (N = 753) in their hotels who were willing to join our research project by completing our questionnaires at two different times. These participants were informed that their responses would be anonymous and kept confidential; that is, they only reported six numbers of their ID cards to help re-searchers match their responses from the two different times. Since we used a time-lagged research design, one of the authors submitted paper-and-pencil questionnaires to these employees during their working hours at two different time points with a time interval of one month. After completing these questionnaires, these employees reported their surveys back to one of the authors who was an assistant in the process of submitting questionnaires in these hotels.” (p. 6, line 259-269).
- The analysis was appropriate, sufficient and well-conducted. But I suggest that a report of the confirmatory factor analysis in a more thorough manner would also provide more strength to the current version.
Response: Thank you for this helpful comment. We ran a CFA to distinguish job insecurity and precariousness. Specifically, we inserted that “
The results revealed multiple distinct factors, with the first unrotated factor accounting for only 39.09% of the total variance extracted. Thus, CMB was not a serious concern in our data. Second, we conducted a confirmatory factor analysis (CFA) for job precariousness, job insecurity, negative affect, and withdrawal behavior to evaluate the discriminant validity of the measure in the present research. As shown in Table 1, alternative models indicated a poor fit to the date—i.e., the three-factor model (χ2 /df = 10864.66/662.00, p < .001, CFI = 0.69, RMSEA = 0.18, IFI = 0.69, TLI = 0.67, RMR = 0.16), the two-factor model (χ2 /df = 15072.68/664.00, p < .001, CFI = 0.56, RMSEA = 0.22, IFI = 0.56, TLI = 0.53, RMR = 0.13), and the one-factor model (χ2 /df = 19581.45/665.00, p < .001, CFI = 0.42, RMSEA = 0.25, IFI = 0.42, TLI = 0.39, RMR = 0.22). Our proposed four-factor model demonstrated a better fit to the data (χ2 /df = 2963.86/623.00, p < .001, CFI = 0.93, RMSEA = 0.08, IFI = 0.93, TLI = 0.92, RMR = 0.06) than all the alternative models. Therefore, the results provided strong support for the distinctiveness of the four study variables for subsequent analyses [55].” (p. 7, line 316-329).
- Under the practical implications section, the authors are suggested to provide literature to support the suggestions to the practical field. Moreover, specific cases to demonstrate the practices from the real world might be a good way to provide much clearer implications to readers.
Response: Thank you for this valuable comment. Through adding examples in the new manuscript, we provided more practical advice to organizations and managers in the modern organizations, especially in the service-oriented organizations. For example, we added that “For example, leaders could set an agenda to give employees performance feedback regularly and provide support for those in need [77]. In doing so, employees are more likely to bring optimism throughout the workplace. Additionally, organizations can use psychometrically sound personality measures to assess critical attributions in hiring and screening processes as well as in employee care because certain traits impact positive or negative motivational states and behaviors [78]. Our findings suggest screening for negative affect to provide targeted care, especially for those who experience high levels of job precariousness.” (p. 12, line 473-480). Please refer to p. 11-12 for detailed argumentations.
7. There are punctuation errors that hinder the quality of this manuscript. I recommend authors to go through the text with a revision perspective or seek aid from a professional editor.
Response: Thank you for your reminders. We have sent our paper to a professional English editor for proofread.

Reviewer 3 Report
I appreciated the opportunity to review this paper. Precariousness is an important topic and important to research.
I identified several concerns about the research and opportunities for clarification. I hope these are helpful.
1) The paper would benefit from being clear about definitions early on. For example, the first two pages of the paper note several constructs omnipresent job precariousness, work precarity, job insecurity, etc. It is unclear how these are related or distinct, or whether these are perceptual or objective.
2) The distinction between precarious work and job insecurity is not clear to me either in theory/definition or in measurement. For example, the sample item for job precariousness is “To what extent are you concerned about losing your current job in the near future?”" The sample item for job insecurity is “I feel uncertain about the future of my job.” These seem to be identical items and typically load together on a scale (or, some researchers consider being concerned about losing one's job to be an outcome of feeling cognitively uncertain about one's job). Moreover, definitions of precarious work often include job insecurity as an element of precarious work.
3) Given the line of reasoning in the paper, it seems that stress would be the best fitting mediator, instead of job insecurity.
4) I wondered why precarious workers would not try to reduce withdrawal behavior in order to reduce uncertainty. In other words, wouldn't vulnerable workers want to try to make their jobs more secure by not engaging in withdrawal behavior?
5) The sampling procedure could be clarified, especially with regard to how the organization provided a list of workers and the consent process. Were surveys anonymous? Were participants compensated? Did they complete the surveys on work time? Were results reported back to the organization?
6) It would be helpful to provide attrition analyses.
7) Similar to my comment #2, it would be helpful to run a CFA to distinguish job insecurity and precariousness.
8) The item for withdrawal behavior looks to be about thinking about withdrawal behavior, rather than actual withdrawal behavior. This distinction is quite important for drawing implications from the results. It would be helpful to clarify and, if only some items in the scale are behavioral, to repeat the analyses with the behavioral items.
9) Looking at the results, the effect of job precariousness strengthens when job insecurity is included. This again raises questions of whether job insecurity can be thought of as a mediator.
More minor notes:
10) Be careful to not editorialize. For example, "
For example, when the novel coronavirus (COVID-19) pandemic spread in 2019, precarious workers, characterized as a marginalized population, were more likely to be laid off or forced to work in unsafe conditions [6] since their feelings are often overlooked [7,8]." There are likely other reasons other than overlooking feelings for what happened to precarious workers.
Author Response
RESPONSES TO REVIEWER 3’S COMMENTS
I appreciated the opportunity to review this paper. Precariousness is an important topic and important to research.
I identified several concerns about the research and opportunities for clarification. I hope these are helpful.
Response: Thank you very much for your positive and encouraging comments on our manuscript. Your insightful guidance has helped us a lot during our revision. We have revised our paper in accordance with your comments and suggestions. We hope that you will find our revision addressing your concerns. Below are our point-by-point responses to your concerns
1) The paper would benefit from being clear about definitions early on. For example, the first two pages of the paper note several constructs omnipresent job precariousness, work precarity, job insecurity, etc. It is unclear how these are related or distinct, or whether these are perceptual or objective.
Response: Thank you for pointing out this issue. It’s a very helpful suggestion. Accordingly, we put the definitions early on (see job precariousness on p. 1, job insecurity on p. 2, and withdrawal behavior on p. 2).
Regarding the constructs of job precariousness, work precarity and job insecurity, they are all perceptual, not objective. In the existing literature, job precariousness (Creed et al., 2020) and work precarity (Allan et al., 2021; Mai et al., 2019) are used interchangeably. To avoid misleading readers, we deleted the construct work precarity and used the construct job precariousness only. As to job insecurity, it’s distinct from job precariousness (see the comparison of these two constructs on p. 2).
2) The distinction between precarious work and job insecurity is not clear to me either in theory/definition or in measurement. For example, the sample item for job precariousness is “To what extent are you concerned about losing your current job in the near future?”" The sample item for job insecurity is “I feel uncertain about the future of my job.” These seem to be identical items and typically load together on a scale (or, some researchers consider being concerned about losing one's job to be an outcome of feeling cognitively uncertain about one's job). Moreover, definitions of precarious work often include job insecurity as an element of precarious work.
Response: Thank you for pointing out. It’s worth noting that job precariousness is, theoretically, different from job insecurity in definition as job insecurity is the perception of potential job loss as a whole, as well as a deterioration in job conditions and a loss of anticipated opportunities (Shoss, 2017), while job precariousness is an individual’s subjective perception of objective job features (uncertainty, low income, and limited social benefits and statutory entitlements (Vosko, 2010)).
Although they are both conceptual, job precariousness is a phenomenon that is present oriented, implying the perception of holding a low-quality job (Kalleberg, 2012), while job insecurity is future oriented, indicating the perception of potential job threatening factors (Shoss, 2017). When developing job precariousness scale, Creed et al. (2020) also provided support for construct validity that job precariousness and job insecurity are different constructs.
Moreover, consistent with your comment 7# below, we conducted a CFA to distinguish job insecurity and precariousness. Specifically, the results (shown in the table below) indicated that two-factor model demonstrated a better fit to the data (χ2 /df = 4860.81/169.00, p < .001, CFI = 0.69, RMSEA = 0.24, IFI = 0.69, TLI = 0.66, SRMR = 0.08) than one-factor model (χ2 /df = 8479.85/170.00, p < .001, CFI = 0.46, RMSEA = 0.32, IFI = 0.46, TLI = 0.40, SRMR =0.26).
Thus, job insecurity and job precariousness are distinctive constructs.
Finally, consistent with Editors’ comment above, we removed the three items and re-run the analysis to test our hypotheses. Consistent with our previous results, all the hypotheses are supported. Please refer to detailed reports and results on p. 7-10.
3) Given the line of reasoning in the paper, it seems that stress would be the best fitting mediator, instead of job insecurity.
Response: Thanks for your comment. Our logic lies in that when employees experience job precariousness, they actually undergo resource shortage. This feeling of resource shortage is not merely present-oriented, but will spread to the foreseeable future and cause a feeling of uncertainty in the future. Thus, employees’ current perception of job precariousness leads to their future-oriented perception of job insecurity. According to conservation of resources (COR) theory, when employees don’t have enough resources at hand to complete their work, they tend to withdraw from work, both psychologically and physically, to restore the resources.
However, just as you pointed, stress is a very interesting and promising perspective, and we can draw on this perspective in the future research, which we added in the limitation section: “In addition, since precarious work reflects not merely temporary features of the business cycle but structural transformations of contemporary employment [28], job precariousness could be a chronic workplace stressor. Therefore, future research could take a stress perspective and examine the long-term effect of job precariousness on other behavior outcomes.” (p. 12).
4) I wondered why precarious workers would not try to reduce withdrawal behavior in order to reduce uncertainty. In other words, wouldn't vulnerable workers want to try to make their jobs more secure by not engaging in withdrawal behavior?
Response: Thank you for your constructive comments. we made the following revisions:
- First, we strengthened our reasonings in the text by claiming that we strictly followed the theoretical framework of conservation of resources (COR) theory to reason our argumentations of the mediating role of job insecurity. Specifically, according to COR, when individuals perceive their precarious job, they develop a sense of resource loss. In this situation, their immediate response tends to display negative behaviors. Thus, we propose that job insecurity contributes to withdrawal behavior.
Meanwhile, previous studies have evidenced that job insecurity causes far-reaching negative outcomes (e.g., psychological ill-being) (Bentzen et al., 2020). As a result, employees are more likely to produce negative behaviors in the workplace (e.g., withdrawal behaviors) (Ryff & Singer, 2007). Consistently, our theoretical argumentations about job precariousness leading to withdrawal behavior via job insecurity can be supported theoretically.
- In addition, we acknowledged your suggestion that vulnerable workers may try to make their jobs more secure by not engaging in withdrawal behavior. Thus, we added this in the limitation section to encourage researcher to explore the possibilities in the future. Specifically, we inserted that “Finally, although we follow COR to propose and test the contribution of job insecurity to employees’ withdrawal behavior, it is possible that vulnerable workers may try to make their jobs more secure by not engaging in withdrawal behavior. Thus, future research is encouraged to explore whether individuals with a certain type of personality would decrease their withdrawal behaviors when they feel the insecurity of their jobs. For example, employees with a proactive personality are more strategic in coping with difficulties and challenges. When they experience job insecurity at some point in the workplace, they are more likely to exhibit crafting behaviors to increase social or structural job resources and challenging job demands and to decrease hindering job demands instead of generating withdrawal behavior.” (p.12, line 503-512).
5) The sampling procedure could be clarified, especially with regard to how the organization provided a list of workers and the consent process. Were surveys anonymous? Were participants compensated? Did they complete the surveys on work time? Were results reported back to the organization?
Response: We appreciate your comment, which motivates us to improve the quality of sampling section. Specifically, we added detailed information about our sampling procedure by inserting that “These HR departments asked all their employees in the hotels with aims and descriptions of our current research project to confirm whether they would participate. Then, the HR departments provided a list of employees (N = 753) in their hotels who were willing to join our research project by completing our questionnaires at two different times. These participants were informed that their responses would be anonymous and kept confidential; that is, they only reported six numbers of their ID cards to help re-searchers match their responses from the two different times. Since we used a time-lagged research design, one of the authors submitted paper-and-pencil questionnaires to these employees during their working hours at two different time points with a time interval of one month. After completing these questionnaires, these employees reported their surveys back to one of the authors who was an assistant in the process of submitting questionnaires in these hotels.” (p. 6, line 259-270).
In addition, since these participants were willing to join our research project, they were not compensated.
6) It would be helpful to provide attrition analyses.
Response: Thank you for your valuable suggestion. We made the following revision in our new manuscript:
First, consistent with your previous comment, we added more descriptions on our sampling. Specifically, we inserted that we informed participants that our sampling is randomly, and only complete surveys can be considered to analyses. Thus, incomplete responses were discarded before further data analyses. That is, the analyses of incomplete responses was not allowed in the current study. This is a reasonable method as statistical research has conceptually suggested that if the characteristics of people lost to follow-up do not differ between the randomized groups, attrition would not introduce bias (e.g., Dumville, Torgerson, & Hewitt, 2006). Therefore, attrition analyses were not applicable to our current study since there was no different characteristics correlated with the participants’ outcome measures.
Second, we acknowledged your valuable suggestion statistically; thus, we added this as a limitation to encourage further research to take the attrition analyses into consideration. In this vein, our current findings can be replicated and generalized.
7) Similar to my comment #2, it would be helpful to run a CFA to distinguish job insecurity and precariousness.
Response: Thanks for this suggestion. We ran a CFA to distinguish job insecurity and precariousness. As the table below shows, two-factor model demonstrated a better fit to the data (χ2 /df = 4860.81/169.00, p < .001, CFI = 0.69, RMSEA = 0.24, IFI = 0.69, TLI = 0.66, SRMR = 0.08) than one-factor model (χ2 /df = 8479.85/170.00, p < .001, CFI = 0.46, RMSEA = 0.32, IFI = 0.46, TLI = 0.40, SRMR =0.26). Therefore, the result shows that job insecurity and job precariousness are distinctive constructs.
Table 1. Confirmatory factor analysis.
Variables |
χ2 |
df |
CFI |
RMSEA |
IFI |
TLI |
SRMR |
Two-factor model |
4860.81 |
169.00 |
0.69 |
0.24 |
0.69 |
0.66 |
0.08 |
One-factor model |
8479.85 |
170.00 |
0.46 |
0.32 |
0.46 |
0.40 |
0.26 |
N=472; Two-factor model: Job precariousness, Job insecurity; One-factor model: Job precariousness + Job insecurity.
8) The item for withdrawal behavior looks to be about thinking about withdrawal behavior, rather than actual withdrawal behavior. This distinction is quite important for drawing implications from the results. It would be helpful to clarify and, if only some items in the scale are behavioral, to repeat the analyses with the behavioral items.
Response: Thanks for this very helpful comment! In the new manuscript, we make the definition and measurement of withdrawal behavior more clearly. In definition, withdrawal behavior refers to a set of attitudes and behaviors used by employees when they maintain a job but for some reason attempt to be less participative (Shapira-Lishchinsky & Even-Zohar, 2011). It represents a sequence of behaviors that begins with episodic psychological withdrawal and, through increasing degrees of withdrawal, gradually expands to physical withdrawal (Lehman & Simpson, 1992) (p. 2).Therefore, withdrawal behavior compasses both psychological and physical withdrawal. In measurement, sample items for psychological withdrawal behavior and physical withdrawal behavior are “How often have you thought of being absent” and “How often have you left work early without permission” respectively. (p. 7). Withdrawal Behavior Scale is developed by Lehman and Simpson (1992), which has good validity and is used widely.
9) Looking at the results, the effect of job precariousness strengthens when job insecurity is included. This again raises questions of whether job insecurity can be thought of as a mediator.
Response: Thanks for this comment. In the previous version, we accidentally miswrote the main effect value before we put in the mediation variable and we reported more details about the mediation effects in the new manuscript. In table 3, we can see that the job precariousness is positively associated with withdrawal behavior (β = 0.63, p < 0.001). After the job insecurity is entered in the regression, the relation between job precariousness and withdrawal behavior is reduced (β = 0.48, p < 0.001), which shows that job insecurity partially mediates the effect of job precariousness on withdrawal behavior. Consistent with our response to your previous comment #3, job insecurity is a mediator theoretically and statistically (p. 8-9).
More minor notes:
10) Be careful to not editorialize. For example, "
For example, when the novel coronavirus (COVID-19) pandemic spread in 2019, precarious workers, characterized as a marginalized population, were more likely to be laid off or forced to work in unsafe conditions [6] since their feelings are often overlooked [7,8]." There are likely other reasons other than overlooking feelings for what happened to precarious workers.
Response: Thank you for your reminder. We are sorry that our previous writing makes the ambiguities. We corrected the errors you identified in our revised version by deleting the sentence.
In sum, we greatly appreciate your constructive feedbacks on our earlier version. Your insightful comments and suggestions have helped us bring out the potential contributions of our study. We hope that our revision has addressed your concerns in a satisfactory manner. Should you find any additional changes necessary, we are more than happy to further revise our paper.
References:
Aiken, L. H., Cimiotti, J. P., Sloane, D. M., Smith, H. L., Flynn, L., & Neff, D. F. (2011). Effects of Nurse Staffing and Nurse Education on Patient Deaths in Hospitals With Different Nurse Work Environments. Medical Care, 49(12), 1047-1053.
Allan, B. A., Autin, K. L., & Wilkins-Yel, K. G. (2021). Precarious work in the 21st century: A psychological perspective. Journal of Vocational Behavior, 126, 103491.
Bentzen, M., Kentt, G., Richter, A., & Lemyre, N. (2020). Impact of job insecurity on psychological welland ill-Being among high performance coaches. International Journal of Environmental Research and Public Health, 17(19).
Burgess, J., Connell, J., & Winterton, J. (2013). Vulnerable workers, precarious work and the role of trade unions and HRM. International Journal of Human Resource Management, 24(22), 4083-4093.
Creed, P. A., Hood, M., Selenko, E., & Bagley, L. (2020). The Development and Initial Validation of a Self-Report Job Precariousness Scale Suitable for Use With Young Adults Who Study and Work. Journal of Career Assessment, 28(4), 636-654.
De Witte, H., & Näswall, K. (2003). ‘Objective’ vs ‘subjective’ job insecurity: Consequences of temporary work for job satisfaction and organizational commitment in four European countries. Economic and Industrial Democracy, 24(2), 149-188.
Dumville, J. C., Torgerson, D. J., & Hewitt, C. E. (2006). Reporting attrition in randomised controlled trials. British Medical Journal, 332(7547), 969–971.
Helbling, L., & Kanji, S. (2018). Job insecurity: Differential effects of subjective and objective measures on life satisfaction trajectories of workers aged 27–30 in Germany. Social Indicators Research, 137(3), 1145-1162.
Kalleberg, A. L. (2012). Job quality and precarious work: Clarifications, controversies, and challenges. Work and Occupations, 39(4), 427-448.
Klandermans, B., Hesselink, J. K., & Van Vuuren, T. (2010). Employment status and job insecurity: On the subjective appraisal of an objective status. Economic and Industrial Democracy, 31(4), 557-577.
Kretsos, L., & Livanos, I. (2016). The extent and determinants of precarious employment in Europe. International Journal of Manpower, 25-43.
Lehman, W. E., & Simpson, D. D. (1992). Employee substance use and on-the-job behaviors. Journal of Applied Psychology, 77(3), 309-321.
Lewchuk, W. (2017). Precarious jobs: Where are they, and how do they affect well-being? The Economic and Labour Relations Review, 28(3), 402-419.
Lewchuk, W., Clarke, M., & De Wolff, A. (2008). Working without commitments: precarious employment and health. Work, Employment and Society, 22(3), 387-406.
Mai, Q. D., Jacobs, A. W., & Schieman, S. (2019). Precarious sleep? Nonstandard work, gender, and sleep disturbance in 31 European countries. Social Science & Medicine, 237, 112424.
Maxwell, G. A., & Grant, K. (2021). Commercial airline pilots' declining professional standing and increasing precarious employment. International Journal of Human Resource Management, 32(7), 1486-1508.
McNamara, M., Bohle, P., & Quinlan, M. (2011). Precarious employment, working hours, work-life conflict and health in hotel work. Applied Ergonomics, 42(2), 225-232.
Mrozowicki, A., Karolak, M., & Krasowska, A. (2016). Between commitment and indifference: Trade unions, young workers and the expansion of precarious employment in Poland. In Labour and Social Transformation in Central and Eastern Europe (pp. 242-260): Routledge.
Perulli, A. (2003). Economically dependent/quasi-subordinate (parasubordinate) employment: legal, social and economic aspects. Study. Brussels: European Commission.
Peugh, J. L., & Enders, C. K. (2004). Missing data in educational research: A review of reporting practices and suggestions for improvement. Review of Educational Research, 74(4), 525-556.
Quinlan, M., & Bohle, P. (2015). Job Quality: The Impact of Work Organisation on Health: The Federation Press.
Ryff, C. D., & Singer, B. (2007). What to do about positive and negative items in studies of psychological well-Being and ill-Being? . Psychotherapy and Psychosomatics, 76(1), 61-62.
Scott, B. A., & Barnes, C. M. (2011). A multilevel field investigation of emotional labor, affect, work withdrawal, and gender. Academy of Management Journal, 54(1), 116-136.
Shapira-Lishchinsky, O., & Even-Zohar, S. (2011). Withdrawal behaviors syndrome: An ethical perspective. Journal of Business Ethics, 103(3), 429-451.
Shoss, M. K. (2017). Job insecurity: An integrative review and agenda for future research. Journal of management, 43(6), 1911-1939.
Smith, V. (2018). Crossing the great divide: Cornell University Press.
Triana, M. D., Trzebiatowski, T., & Byun, S. Y. (2017). Lowering the Threshold for Feeling Mistreated: Perceived Overqualification Moderates the Effects of Perceived Age Discrimination on Job Withdrawal and Somatic Symptoms. Human Resource Management, 56(6), 979-994.
Vosko, L. F. (2010). Managing the margins: Gender, citizenship, and the international regulation of precarious employment: Oxford University Press.
Wang, H. J., Lu, C. Q., & Lu, L. (2014). Do people with traditional values suffer more from job insecurity? The moderating effects of traditionality. European Journal of Work and Organizational Psychology, 23(1), 107-117.

Round 2
Reviewer 3 Report
I appreciate the authors' thorough responses and think these have helped the paper.
I was confused by the response to the question about attrition. For attrition analyses, you could compare the people who only responded to one survey and the people who responses to both surveys on the demographic variables and on the main study variables. In the response, the authors' note "statistical research has conceptually suggested that if the characteristics of people lost to follow-up do not differ between the randomized groups, attrition would not introduce bias." This requires determining whether there are sample differences between those who remained and those who dropped out of the study.
I appreciate more detail about the sample. However, I still have some ethical concerns here given that the HR department knows who agreed to participate.
Author Response
RESPONSES TO REVIEWER 3
I appreciate the authors' thorough responses and think these have helped the paper.
Response: Thank you for your positive feedback and constructive comments on our revised manuscript. We really appreciate that your comments and suggestions help us a lot. We have revised our paper in accordance with your comments and suggestions. Below are our point-to-point responses to your concerns.
I was confused by the response to the question about attrition. For attrition analyses, you could compare the people who only responded to one survey and the people who responses to both surveys on the demographic variables and on the main study variables. In the response, the authors' note "statistical research has conceptually suggested that if the characteristics of people lost to follow-up do not differ between the randomized groups, attrition would not introduce bias." This requires determining whether there are sample differences between those who remained and those who dropped out of the study.
Response: Thank you very much for this insightful comment. In our sampling procedure, we strictly used the random sampling where each participant has an equal chance of being selected without sampling bias. Thus, it could be reasonable to argue that there are no sample differences between those who remained and those who dropped out of the study.
We admitted that given our assumption of no differences (as mentioned above), we did not take the sample who dropped out of the study into our consideration for data analyses. That is, we directly deleted the samples of incomplete responses. As a result, we failed to conduct the attrition analyses.
However, we indeed acknowledged your valuable point about the potential differences between those who remained and those who dropped out of the study. Therefore, in the new manuscript, we added more argumentations in the limitation section to encourage scholars in the future to replicate our findings with conducting a more rigor research design and considering the attrition analyses. Please refer to p. 12 (line 506-512) for our detailed argumentations.
I appreciate more detail about the sample. However, I still have some ethical concerns here given that the HR department knows who agreed to participate.
Response: Thank you for pointing this. We are sorry for our unclear explanations in the sampling section that make you confused. In the new manuscript, we added more details about our sampling to explicitly reveal the process of assistance from HR departments. Specifically, the HR departments first helped at the beginning of our survey submission; that is, before survey submission, the HR departments asked all the employees in their hotels to confirm that they (i.e., employees) were willing to participate in our research project by filling the questionnaires at two different points of time. After receiving these employees’ responses (N = 753), the HR departments provided a list of these participants with six numbers of their ID cards and their telephone numbers. These employees’ ID information can be tracked by the employee files in the HR departments. When we submit our questionnaires, our research team used the list of the ID information to submit our questionnaires to these participants by giving them a phone call. Thus, these employees without telling us their names could fill in the questionnaires by reporting the six numbers of their ID cards. Please refer to our corrections in the new manuscript on p. 6.
We really appreciate your insightful and constructive comments and they help us a lot in revising our manuscript. We hope you will like our responses and new version!
